# Balancing Tumor Immunotherapy and Immune-Related Adverse Events: Unveiling the Key Regulators

**DOI:** 10.3390/ijms252010919

**Published:** 2024-10-10

**Authors:** Jianshang Huang, Lei Xiong, Sainan Tang, Junhao Zhao, Li Zuo

**Affiliations:** 1Laboratory of Molecular Biology, Department of Biochemistry, School of Basic Medical Sciences, Anhui Medical University, No.81, Meishan Rd., Hefei 230032, China; huangjianshang111@163.com (J.H.);; 2Innovation and Entrepreneurship Laboratory for College Students, Anhui Medical University, No.81, Meishan Rd., Hefei 230032, China

**Keywords:** irAEs, ICI colitis, CTLA4, PD-1/PD-L1, MLCK, FKBP8

## Abstract

Tumor immunotherapy has emerged as a promising approach in cancer treatment in recent years, offering vast potential. This method primarily involves targeting and inhibiting the suppressive checkpoints present in different immune cells to enhance their activation, ultimately leading to tumor regression. However, tumor cells exploit the surrounding immune cells and tissues to establish a tumor microenvironment (TME) that supports their survival and growth. Within the TME, the efficacy of effector immune cells is compromised, as tumor cells exploit inhibitory immune cells to suppress their function. Furthermore, certain immune cells can be co-opted by tumor cells to facilitate tumor growth. While significantly enhancing the body’s tumor immunity can lead to tumor regression, it can also result in severe toxic side effects and an inflammatory factor storm. As a consequence, patients often discontinue treatment due to immune-related adverse events (irAEs) or, in extreme cases, succumb to toxic side effects before experiencing tumor regression. In this analysis, we examined several remission regimens for irAEs, each with its own drawbacks, including toxic side effects or suppression of tumor immunotherapy, which is undesirable. A recent research study, specifically aimed at downregulating intestinal epithelial barrier permeability, has shown promising results in reducing the severity of inflammatory bowel disease (IBD) while preserving immune function. This approach effectively reduces the severity of IBD without compromising the levels of TNF-α and IFN-γ, which are crucial for maintaining the efficacy of tumor immunotherapy. Based on the substantial similarities between IBD and ICI colitis (combo immune checkpoint inhibitors-induced colitis), this review proposes that targeting epithelial cells represents a crucial research direction for mitigating irAEs in the future.

## 1. Introduction

In recent years, there has been a notable increase in the incidence of cancer [1]. Unfortunately, the conventional treatment approaches for cancer, such as surgery in combination with chemotherapy, radiotherapy, and targeted therapy, often fall short of achieving satisfactory outcomes [2]. However, the emergence of tumor immunotherapy has revolutionized cancer treatment [3]. One of the most prevalent forms of tumor immunotherapy is immune checkpoint blockade (ICB), which offers significant benefits to patients with advanced cancers, including melanoma [4], non-small cell lung cancer [5], renal cell cancer [6], and certain colorectal cancer patients with mismatch repair defects [4,7,8].

However, immune checkpoint blockade therapy often leads to varying degrees of side effects, including enteritis [9], hepatitis [10], dermatitis [10], and pneumonia [11]. Among these, severe enteritis and hepatitis are more frequently observed [9,10], and these side effects are commonly referred to as irAEs [12]. Patients often experience treatment interruptions due to severe toxic side effects, which can impact the effectiveness of tumor treatment [13,14,15].

This review explores the intricate relationship between tumor immunotherapy and the TME. It provides an overview of how immunotherapy modulates the TME and discusses its impact on immune evasion and treatment resistance. Moreover, it examines the advantages of immunotherapy in this context. The review also addresses the common side effects of tumor immunotherapy and their effects on the immune system. Lastly, it emphasizes the importance of targeting epithelial cells to mitigate irAEs.

## 2. Tumor Immunotherapy

### 2.1. Origin of Tumor Immunotherapy

The surgeon William Coley is widely regarded as the “father of tumor immunotherapy.” In 1893, Dr. Coley made a significant discovery when he observed that a patient’s tumor could reappear after surgical removal [16]. During the patient’s prolonged battle with the tumor, they experienced bacterial infections and recurrent episodes of high fever. Remarkably, on several occasions following these fever episodes resulting from the bacterial infection, the tumor exhibited shrinkage or even regression. Building upon this observation, Dr. William Coley pioneered a novel approach to treat tumors by administering patients with a dose of lipopolysaccharide (LPS) to stimulate the immune system. This marked the first recorded instance of tumor immunotherapy. Subsequently, there was a significant lull in the development of tumor immunotherapy until the breakthrough discoveries of anti-CTLA4 (cytotoxic T-lymphocyte-associated protein 4) in 1987 [17] and anti-PD1 (programmed cell death protein 1) in 1992 [18]. These discoveries ushered in a new era in tumor immunotherapy research and paved the way for substantial advancements in the field of tumor immunotherapy.

### 2.2. Current Strategies for Tumor Immunotherapy

Tumor immunology is primarily focused on investigating tumor immunogenicity, immune responses, and immune status, as well as the occurrence, development, diagnosis, and treatment of tumors. Historically, tumor immunotherapy was primarily utilized as an adjunct therapy alongside surgery, chemotherapy, and radiotherapy, aiming to enhance the body’s immune system function to control and eliminate certain tumor cells. However, recent advancements in tumor immunotherapy, such as immune checkpoint blockade therapy [19], tumor vaccines [20], CAR-T cell therapy [21] (where CTL cells are extracted from patients, genetically modified to improve their ability to recognize and kill tumor cells, expanded in vitro, and then reintroduced into patients), as well as CAR-NK therapy [22] and other tumor immune approaches, have propelled tumor immunotherapy into the forefront as a mainstream treatment option.

Immune checkpoint blockade has demonstrated enduring tumor regression in the treatment of advanced human cancers [19]. The primary objective of tumor immunotherapy is to enhance its therapeutic efficacy. To achieve this goal, several strategies can be employed. Firstly, inhibiting immune checkpoints can eliminate tumor-associated immune cells. Moreover, mitigating the harmful effects of the inflammatory cytokine storm on normal tissues, such as ICI colitis, graft-versus-host disease (GVHD), and inflammatory liver disease, is crucial. Furthermore, disrupting the mechanisms by which tumor cells evade immune therapy or inhibiting the suppressive functions of immune cells is of great importance in enhancing the efficacy of tumor immunotherapy. These collective approaches significantly contribute to the reinforcement of tumor immunotherapy. Currently, immune checkpoint blockade represents the most prevalent and effective form of tumor immunotherapy.

The fundamental principle of immune checkpoint blockade therapy is to obstruct the receptors that restrain immune cells, thereby activating immune cells that are in a quiescent state. This activation aims to strengthen or stimulate the patient’s immune system to effectively eliminate tumor cells.

Tumor cells frequently exhibit a significant abundance of specific antigens [23] or non-coding RNA [24], which contribute to modifying the TME and influencing various immune cells. Simultaneously, cancer, being a chronic process, often leads to immune cell depletion, resulting in cancer patients frequently experiencing immune surveillance evasion or immune cell exhaustion [25,26]. Consequently, the immune system of cancer patients often finds itself in a state where it is unable to recognize tumor cells or lacks the ability to effectively eliminate tumor cells due to insufficient immune cells or weakened immune function.

Various immune checkpoint blockades have the potential to activate a majority of tumor immune cells. By targeting immunosuppressive molecules such as PD-1/PD-L1 [27], CTLA-4 [28], TIM-3 [29], and NKG2A [30], or combining them with tumor-specific antigens, they can activate immune cells and enable them to effectively target the TME, resulting in specific tumor cell-killing effects. This approach not only alleviates tumor-related symptoms but also slows down disease progression, leading to favorable prognoses.

Unlike surgery, radiotherapy, and chemotherapy, which may not completely eliminate hidden tumor cells [31], immune checkpoint blockade therapy can yield durable and more comprehensive tumor cell elimination effects. As a result, tumor immunotherapy possesses distinct advantages over traditional cancer treatments. The primary objective of immune checkpoint blockade is to activate specific immune cells by selectively blocking inhibitory targets on various immune cells, thus facilitating the eradication of tumor cells.

Antigen-presenting cells (APCs) express ligands known as B7 (including CD80/86), which can bind to the CD28 receptors on T cells, providing an activation signal. Additionally, APCs express major histocompatibility complex class II (MHC-II) molecules that bind to the T cell receptor (TCR), allowing for antigen recognition (Figure 1A,B). The combined interaction of these signal pairs stimulates the activation of T cells. However, CTLA-4, which acts as an inhibitory signal, can competitively inhibit CD28 molecules and bind to B7 molecules, thereby hindering TCR signaling and suppressing T cell activation. This inhibition occurs because CTLA-4 molecules exhibit a higher affinity for B7 molecules [32,33]. Consequently, targeted blockade of CTLA-4 can effectively activate various T lymphocytes.

PD-1 is primarily expressed on cytotoxic T lymphocytes (CTLs) and serves as an inhibitor of cell activity. On the other hand, PD-L1 is mainly expressed in tumor cells and suppressor immune cells. CTLs are responsible for eliminating foreign or abnormal cells (Figure 1A). When PD-1 binds to PD-L1, it leads to the inhibition of CTL cell function, facilitating the evasion of tumor immunity or immune surveillance [34]. By blocking either PD-1 or PD-L1, the function and activity of CTL cells can be significantly enhanced.

Currently, the most prevalent immune checkpoint blockade approach in clinical tumor immunotherapy involves the use of anti-CTLA-4 or anti-PD-1/PD-L1 antibodies, either individually or in combination [35,36]. While there are several other immune cell immune checkpoint blockers available, the combination of anti-CTLA-4 and anti-PD-1/PD-L1 has emerged as the most mature and widely adopted method [37,38,39]. This combined approach is commonly referred to as combined immune checkpoint blockade or immune checkpoint inhibitors (CICB or ICIs).

## 3. Immunotherapy Target Cells

### 3.1. CD8^+^ T Cells

Effector CD8^+^ cells are widely recognized as the primary anti-tumor cells within the body due to their potent anti-tumor capabilities. These CD8^+^ cells predominantly target tumor cells by recognizing the major histocompatibility complex class I (MHC-I) molecules, referred to as human leukocyte antigen (HLA) in humans, present on the surface of tumor cells, thereby exerting their anti-tumor effects. A particular study [40] conducted on non-small cell lung cancer (NSCLC) patients observed a notable upregulation of PD-1/PD-L1 expression alongside a downregulation of MHC-I expression within the tumor tissues. Furthermore, the HLA-I^+^/PD-L1^+^ tumor tissues exhibited a high level of infiltration by CD8^+^ T lymphocytes. Conversely, tumor tissues lacking HLA-I showed a significant reduction in the number of tumor-infiltrating lymphocytes (TILs).

Effector CD8^+^ T cells, commonly referred to as CTLs, primarily exert their anti-tumor effects through the release of perforin and granzyme B(GZMB), which mediate tumor cell lysis and apoptosis. Moreover, activated effector CD8^+^ T cells can induce tumor cell apoptosis via the classical FASL–FAS pathway (Figure 1B). Additionally, these effector CD8^+^ T cells are capable of releasing cytokines such as IFN-γ [40], TNF-α [41], and lymphotoxin-α (LT-α) [42,43] which contribute to the killing and weakening of tumor cells. 

### 3.2. CD4^+^ T Cells

When a tumor cell undergoes cell death, it releases specific tumor antigens that are specifically recognized by APCs such as dendritic cells (DCs) and macrophages. These APCs capture the tumor antigens and process them into MHC-II molecules. MHC-II molecules are then presented on the surface of APCs and can be recognized by various CD4^+^ T helper (TH) cells, including TH1, TH2, and TH17 cells. The interaction between MHC-II and CD28 molecules on CD4^+^ TH cells stimulates their activation. Consequently, activated CD4^+^ TH cells release a range of cytokines, including IFN-γ, TNF-α, IL-1β, and IL-2, which collectively contribute to the anti-tumor effects [41] (Figure 1A,B). After that, the cytokines released by CD4^+^ TH cells can indirectly or directly activate M1 macrophages, natural killer (NK) cells, effector CD8^+^ T cells [41,44,45,46] B cells [41], NKT cells [47], CD4^+^ cytotoxic T lymphocytes [44], and innate lymphoid cells (ILCs) [41]. This activation leads to a synergistic killing effect on tumor cells, further enhancing the anti-tumor immune response.

### 3.3. M1 Type Macrophages

M1 macrophages play a vital role in innate tumor immunity. As part of the innate immune cell repertoire, M1 macrophages exhibit selectivity towards tumor and tumor-associated cells, sparing healthy tissue structures. These macrophages exert their anti-tumor effects through direct or indirect cytotoxicity. Serving as APCs, M1 macrophages possess the ability to process and present antigens via MHC-II molecules to CD4^+^ T helper (TH) cells [48]. Furthermore, M1 macrophages can phagocytose tumor cells directly. By binding to the FC region on the surface of tumor cells through Fc-receptors (FCRs), M1 macrophages initiate the antibody-dependent cell-mediated cytotoxicity (ADCC) pathway, leading to tumor cell elimination [48].

Moreover, activated M1 macrophages release a repertoire of cytokines, including TNF-α [48], IFN-γ [49], IL-2 [50], IL-12/IL-23 [48], and IL-1β [48] which collectively contribute to promoting the immune response against tumors within the human body.

### 3.4. NK Cells

Natural killer cells (NK cells) are widely recognized for their diverse effector functions, resembling those of CTLs and macrophages. They are distributed throughout the body. Similar to CTLs, NK cells typically spare normal tissues from cytotoxic effects. NK cells recognize autologous or allogeneic cells through the MHC-I complex [30], enabling them to eliminate tumor cells through various pathways, including the classical perforin–granzyme pathway, Fas–FasL pathway, ADCC pathway, and TNF-α–IFN-γ pathway [50].

The mechanism underlying the killing of tumor cells by NK cells is rooted in the body’s innate immunity. NK cells possess both killer activating receptors (KARs) and killer inhibitory receptors (KIRs). Tumor tissues exhibit distinct MHC composition compared to normal tissues. The MHC in tumor tissues is referred to as allogeneic MHC, while the MHC in the body is autologous. When KIR fails to recognize the autologous MHC complex and KAR interacts with its ligand, NK cells become activated to eliminate tumor cells. On the other hand, when KIR recognizes autologous MHC-I, it inhibits the cytotoxic activity of NK cells. However, in many cases, the allogeneic MHC-I expressed by tumor cells closely resembles the autologous MHC-I. Consequently, NK cells are unable to distinguish between them. Moreover, certain substances present in the TME can upregulate the expression of specific inhibitory receptors such as NKG2A on NK cells [51], leading to the suppression of NK cell activation.

In a recent study [51], it was observed that tumor tissues exhibited a significant decrease in autologous MHC expression, while there was an increased expression of NKG2A. To investigate the impact of these findings, the researchers employed monalizumab, a monoclonal antibody that blocks NKG2A-mediated recognition of HLA. They also utilized cetuximab, which effectively blocks CD16-mediated recognition of EGFR. These two antibodies were combined in the study. Promising interim results from a phase II trial conducted on patients with previously treated head and neck squamous cell carcinoma demonstrated an objective response rate of 31%. Furthermore, the same group of researchers explored the combination of monalizumab with durvalumab, a monoclonal antibody that hinders the binding of PD-1/PD-L1. This combination proved to be more effective in activating CTL and NK cells, offering patients a more favorable prognosis.

### 3.5. Other Cells

#### 3.5.1. DCs 

DCs function as APCs in the immune system. They can be categorized into four subsets based on the expression of CD11b and CD103 markers. CD11b^(+/−)^CD103^(+/−)^ DC subsets have distinct roles in the immune response. 

A study [52] demonstrated that CD11b^−^CD103^+^ DC cells can directly interact with proinflammatory CD11b^+^CD103^−^ DC cells by upregulating retinoic acid (RA). CD11b^+^CD103^−^ DC cells and CD11b^+^CD103^+^ cells release inflammatory cytokines such as IL-12/23 and IL-1β to act on the TME [48]. Simultaneously, within this microenvironment, CD11b^-^CD103^+^ DC cells produce RA, which interacts with regulatory T (Treg) cells. Treg cells produce various anti-inflammatory factors, primarily acting on Th17 cells and CD11b^+^CD103^−^ DC cells to inhibit inflammation [52]. During this process, the CD137 molecule serves as an immune checkpoint for proinflammatory DC cells. Blocking CD137 can enhance the proinflammatory response of inflammatory DC cells [49].

#### 3.5.2. γδT Cells

In contrast to αβT cells, γδT cells are characterized by the presence of a TCR composed of a γ chain and a δ chain. Similar to NK cells and M1 macrophages, they are categorized as components of the body’s innate immune system. The primary mechanism by which γδT cells eradicate tumors shares similarities with CTL and NK cells, involving pathways such as granzyme–perforin, FAS–FASL, ADCC, and pro-inflammatory cytokines. Diverging from CTLs, γδT cells possess the capacity to recognize tumor antigens independently of MHC restrictions. This distinctive attribute renders γδT cell-based immunotherapy advantageous, particularly in the treatment of hematological and solid tumors, notably leukemia and liver cancer [53]. A previous study [53] demonstrated that tyrosine kinase inhibitors (TKIs), which are commonly utilized as a first-line therapy for leukemia, significantly enhance the abundance and cellular response of γδT cells. Consequently, the development of CAR-γδ T cells holds promising therapeutic prospects for enhanced efficacy in cancer treatment.

#### 3.5.3. B Cells

CD22 is a prominent negative regulator of B cell response, which exerts its inhibitory effect by interacting with CD22L, thereby suppressing the signaling generated by the B cell receptor (BCR) complex during antigen recognition. Blocking CD22 can significantly enhance the anti-tumor response of effector B cells [54]. B cells possess the capability to eliminate tumor cells directly through pathways such as ADCC or phagocytosis. Additionally, they can stimulate T cell immune responses by presenting antigens to T cells via DCs. Moreover, B cells function as dedicated APCs themselves, facilitating T cell activation [55,56].

However, a study [57] has revealed that B cell infiltration is linked to a poor prognosis in renal cell carcinoma and glioblastoma. In many instances, tumor-infiltrating B cells undergo transformation into regulatory B (Breg) cells and secrete cytokines such as IL-10 and TGF-β, which promote tumor progression. Consequently, the application of B cells in tumor immunotherapy remains a complex matter.

## 4. Suppressive Immune Cells That Downregulate Tumor Immune Efficacy

### 4.1. Tumor Microenvironment and Treg Cells

The occurrence and elimination of tumor cells are ongoing processes within the body. In the battle between the immune system and tumor cells, if the rate of tumor cell generation is lower than the rate at which the immune system eliminates them, the tumor cells will either completely perish or enter a latent state. Conversely, if the growth rate exceeds the immune system’s killing rate, the tumor progresses. In cases where both rates are balanced, a seemingly benign tumor typically forms, although tumor cells can be highly deceptive during this stage. Many malignant tumors undergo a latent phase lasting years or even decades before undergoing rapid growth and dissemination. Throughout this process, the TME plays a crucial role [2].

Tumor cells employ various strategies to evade immune surveillance. They can hinder or counteract tumor immune cells through the TME, manipulate the physicochemical state of the metabolic environment, promote the presence of immunosuppressive cells, and exploit cytokines that facilitate tissue growth. These mechanisms enable tumor cells to escape immune recognition and elimination.

The TME is a complex structure consisting of tumor cells, host cells, metabolites, and extracellular matrix components (Figure 2). Typically, it exhibits poor vascularization, resulting in reduced efficiency of nutrient and oxygen supply and inadequate removal of metabolic waste [58]. Consequently, the TME is characterized by high levels of lactic acid accumulation [59]. The hypoxic conditions and limited oxygen utilization capacity lead to increased glycolysis and lactic acid production in various cells, particularly tumor cells, a phenomenon known as the Warburg effect [60]. Tumor cells metabolize glucose to lactate approximately ten times more efficiently than normal tissue.

Under these circumstances, monocarboxylate transporter 1 (MCT1) promotes PD-1 expression in various immune cells [59], especially Treg cells and CTLs. Inhibition of CTL activity and enhanced Treg activity result in increased production of transforming growth factor-beta (TGF-β) and interleukin-10 (IL-10) [59]. TGF-β plays a pivotal role in polarizing M1 macrophages into M2 macrophages, promoting angiogenesis, nutrient supply, and tumor growth. Consequently, this leads to an environment conducive to rapid tumor growth, spread, and metastasis. Targeted blockade of MCT1, PD-1/PD-L1, and TGF-β may partially reverse this process (Figure 2). 

Interestingly, a study [61] on the origin of TGF-β revealed that Treg cells expressing the β8 chain of αvβ8 integrin (Itgβ8) are the primary activators of TGF-β in the TME. αvβ8, produced by cancer cells and stored within the TME, promotes TGF-β expression in Treg cells. Researchers achieved significant inhibition of tumor growth by specifically knocking down Itgβ8 in Treg cells.

### 4.2. M2-Type Macrophages

The role of macrophages in the TME is intriguing. Unlike other immune cells, certain macrophages, known as tumor-associated macrophages (TAMs), undergo polarization toward an M2 phenotype within the TME, often facilitated by interactions with components such as TGF-β [62] (Figure 2). M2 macrophages not only fail to eliminate tumor cells but also actively promote their growth and enhance their survival [63]. They contribute to the development of fibroblasts in the TME, stimulate tumor tissue angiogenesis, induce epithelial–mesenchymal transition (EMT) in tissues, and contribute to the loss of E-cadherin [64]. Moreover, M2 macrophages inhibit the function of IFN-γ^+^CD8^+^ T cells [65], thereby promoting the migration and metastasis of tumor cells and ultimately leading to immune evasion by tumor cells [66]. 

A study investigating osteosarcoma [67] revealed that immune checkpoint molecules associated with M1 macrophages were significantly enriched in the TME, while M2 macrophages were markedly upregulated. Therefore, finding ways to prevent the polarization of M1 macrophages into M2 macrophages within the TME is an important area of research. Strategies aiming to modulate macrophage polarization and promote an anti-tumor M1 phenotype could hold therapeutic potential in combatting tumor growth and metastasis.

Following the targeted inhibition of CTLA4 and PD-1/PD-L1, there is a significant enrichment of TH1 cells and CTL cells, which subsequently produce IFN-γ and immune complexes (formed by the binding of antibodies and antigens). M1 macrophages, upon exposure to one or more immune complexes, LPS, or IL-1β, gradually transition into M2b macrophages [68,69]. M2b macrophages release a small amount of inflammatory factors such as TNF-α and IL-6 but produce a substantial quantity of IL-10, leading to a potent immunosuppressive effect [68,70]. Furthermore, M2b macrophages can further transform into M2c macrophages in the presence of self-released IL-10. M2c macrophages release IL-10 and TGF-β and exhibit strong fibrogenic and angiogenic properties [69]. This phenomenon may explain why immune checkpoint inhibitor (ICI) therapy alone does not yield favorable outcomes in cancer treatment [71].

The immunosuppressive effects mediated by the M2b and M2c macrophages, along with the production of IL-10 and TGF-β, contribute to tumor progression and immune evasion. This highlights the complexity of the TME and the need for comprehensive therapeutic approaches that target multiple immunosuppressive pathways in conjunction with ICI therapy to improve cancer prognosis.

### 4.3. Myeloid-Derived Suppressor Cells 

Furthermore, the TME is enriched with various chemokines, including CCL2 and CXCL3. CCL2 exhibits its effects on myeloid progenitor cells and facilitates the recruitment of differentiated immune cells such as macrophages [72], DC cells [73], myeloid-derived suppressor cells (MDSCs) [74], and mast cells [75], among others. Moreover, CCL2 contributes to the prolonged survival of the recruited cells. For instance, studies have demonstrated that CCL2, also known as monocyte chemoattractant protein (MCP1), binds to CCR2 and recruits cells such as macrophages and DC cells [76]. Upon recruitment to the TME, different DC cells undergo polarization towards an anti-inflammatory phenotype, promoting tumor progression.

One of the primary mechanisms by which CCL2 promotes tumor progression involves its interaction with MDSCs. Research has revealed that tumor cells can evade immune surveillance through various means, including disruption of genes involved in antigen processing and presentation, as well as upregulation of inhibitory immune checkpoint genes. In mouse tumor models, CRISPR screening was performed, revealing a significant loss of tumor suppressor genes (TSGs) and a notable enrichment of CCL2/MCP1 genes, along with an accumulation of MDSCs [74].

MDSCs can be classified into two main subsets: polymorphonuclear PMN-MDSCs and monocytic mo-MDSCs. Within the TME, PMN-MDSCs exert detrimental effects on various T cell populations, particularly CTLs, by upregulating the expression of S100A8/A9 proteins to enhance the generation of reactive oxygen species (ROS) [77]. On the other hand, mo-MDSCs inhibit T cell function by increasing the production of inducible nitric oxide synthase (iNOS) and subsequent nitric oxide (NO) release [77].

Interestingly, MDSCs not only exhibit suppressive effects on T cells but also play a role in promoting Th1-type inflammation. A study discovered that the knockout of RNF5, a gene responsible for ubiquitinating S100A8 protein, resulted in an elevation of CCL2 levels, leading to enhanced recruitment of DCs into the gut and a slight enrichment of CD4^+^ cells [78]. Consequently, even low doses of dextran sulfate sodium (DSS) induced severe IBD due to the promotion of DC accumulation in the gut by S100A8. In this context, DCs significantly enhance the function of effector CD4^+^ cells, resulting in the development of intense Th1-type IBD. However, when this process occurs within the TME, the functionality of DCs undergoes a complete reversal, as they polarize into anti-inflammatory DCs that promote tumor progression.

Overall, the primary role of MDSCs is to suppress inflammation. In a study exploring IBD treatment [79], increasing the proportion of monocytic myeloid-derived suppressor cells (M-MDSCs) in the colon lamina propria led to a significant inhibition of colitis symptoms.

Hence, theoretically, targeting the CCL2(MCP1)/CCR2/MDSC pathway could potentially serve as an immune target for refractory colorectal cancer (CRC). Two studies investigating the function [77] and treatment [71] of MDSC in CRC have revealed elevated levels of CCL2 in the tumors of patients with colitis-associated CRC. CCL2 promotes the accumulation of MDSCs within colon tumors and enhances the immunosuppressive function of polymorphonuclear PMN-MDSCs. Consequently, blocking CCL2 can alleviate tumor progression. However, tumor cells possess intricate mechanisms, including the recruitment of alternative MDSC-inducing chemokines such as CXCL3/CXCR. In a particular study [71], anti-PD1 antibody demonstrated limited efficacy in CRC treatment. Nevertheless, targeted blockade of the IRF2 (Type II interferon regulatory factor), enriched for Type II interferons such as IFN-γ suppressor and Kras (classic sarcoma gene), in combination with anti-PD1, significantly reduced CXCL3-CXCR2 levels and MDSC enrichment in various CRC tumor types, resulting in notable tumor suppression.

In conclusion, MDSCs possess both significant anti-inflammatory effects and tumor-promoting effects, making them critical immune cells exploited by tumor cells. Targeting the MDSC-related CCL2(MCP1)–CCR2 pathway holds promise as a therapeutic approach, particularly in refractory colorectal cancer.

### 4.4. Mast Cells

Mast cells (MCs) are primarily found surrounding visceral microvessels and they have the capacity to release classical pro-angiogenic factors such as VEGF, FGF-2, PDGF, and IL-6 [80]. When tumor tissue becomes vascularized, it experiences accelerated growth due to an enhanced nutrient supply. MCs are enriched in various cancers, including melanoma, breast cancer, and colorectal cancer. MCs actively promote angiogenesis and induce neovascularization through the release of trypsin and chymotrypsin, thereby contributing to the remodeling of the TME [81].

In a study investigating gastric cancer [82], researchers observed that blocking the activator receptor of MCs significantly inhibited the enrichment of MC cells and downregulated the chemokines CCL2/CCL3 in macrophages, resulting in a reduction in gastric cancer cell levels. Targeting MCs can also be combined with ICB for tumor treatment. In a study focusing on melanoma immune tolerance [83], researchers discovered that the efficacy of anti-PD-1 alone was suboptimal. To address this, they constructed a melanoma mouse model and administered anti-PD-1 treatment, which revealed a high abundance of Treg cells and enriched MCs in the TME. Moreover, they found that patients injected with anti-PD-1 exhibited downregulation of HLA-I expression, while the expression of CTL and GZMB was not significantly different from that of normal individuals, potentially contributing to tumor immune tolerance. Subsequently, the researchers administered sunitinib or imatinib in combination with anti-PD-1 drugs, resulting in a significant reduction in MC expression and nearly complete tumor regression.

In conclusion, targeting MCs shows promise as a therapeutic strategy in cancer treatment. Controlling MC activity can impact tumor angiogenesis and the remodeling of the TME, and enhance the efficacy of immunotherapy, such as immune checkpoint blockade.

## 5. Tumor Immunotherapy and irAEs 

### 5.1. The Epidemiology of irAEs

The interaction between tumor immunity and tumor cells within the body can be likened to a battle in a city. Notably, the intestines [9,84] and liver [85], being the largest immune organs and crucial metabolic organs in the human body, often bear the brunt of collateral damage during tumor immunotherapy. IrAEs occurring in various organs are typically categorized into four grades (G1–G4), based on severity. Grade 1 signifies mild adverse events that do not require specific treatment, while grade 2 denotes moderate adverse events necessitating intervention. Severe adverse events requiring hospitalization are classified as grade 3, and life-threatening events fall under grade 4. This grading system aids in evaluating the seriousness of irAEs. Vigilant monitoring and prompt management are essential to mitigate the impact of irAEs and optimize patient outcomes.

The incidence of the most common irAEs, such as dermatitis, approaches nearly 50%. However, these events typically fall within the mild to moderate range (G1/G2 grades) and do not pose a life-threatening risk to patients [86]. On the other hand, severe irAEs often manifest as G3/G4 grade diarrhea, IBD, and hepatitis. A clinical epidemiological meta-analysis of irAEs related to ICB revealed that the probability of severe diarrhea was 9.2%, IBD 13.6% (with 9.1% classified as G3/G4) [87], and inflammatory liver disease 17.6% [88,89]. Internationally, 613 fatal ICI toxic events were reported from 2009 through January 2018 by Vigilyze [88]. The spectrum varied significantly by regimen: of 193 anti-CTLA-4 deaths, 70% (135 cases) were attributed to colitis [88]. Additionally, tumor immunotherapy may lead to myocarditis [88] and increase the risk of type I diabetes [90,91] in patients with thyroid disease [91]. Nevertheless, a significant proportion of severe irAEs are associated with the intestine or liver. Firstly, as the largest immune organ, the intestinal cavity harbors a substantial number of immune cells. Moreover, as a distinct organ connecting the internal and external environments of the human body, the intestinal cavity constantly faces significant external challenges. The complex physical and chemical environment within the intestinal lumen fosters the presence of a diverse array of microorganisms and various metabolites. Consequently, the intestinal cavity is frequently enriched with immune cells associated with inflammation [92]. Similarly, the liver, being a central immune organ, houses a considerable population of innate immune cells such as macrophages, NK cells, and granulocytes. Furthermore, the liver serves as a vital organ for digestion, metabolism, and blood storage, which predisposes it to accumulate diverse metabolites and harmful substances, while also being exposed to metabolites and endotoxins from the intestinal microbiota [93]. Consequently, the immune environment within the liver is highly susceptible to disturbances following ICB treatment. Notably, liver disease often exhibits a close relationship with the condition of the intestine, known as the gut–liver axis. In theory, targeted therapies capable of alleviating intestinal irAEs may indirectly alleviate hepatic irAEs as well.

Anti-CTLA-4 and anti-PD1/PDL1 are the most commonly used ICB biologics in clinical tumor immunotherapy, but they are also associated with significant toxicities and side effects. A systematic review [94] revealed that the incidence of IBD in cancer patients treated with anti-CTLA-4 alone ranged from 8.4% to 11.3% [95]. In contrast, the incidence of IBD in patients treated with anti-PD1/PDL1 monotherapy ranged from 0.3% to 3.4% [87,94,96,97]. Notably, when both anti-CTLA-4 and anti-PD1 were used in combination, the incidence of IBD increased to 14% [94,98]. Additionally, approximately 1% of patients experienced ICB-related enterocolitis complications resulting in death [35,99]. These findings suggest that the combination of anti-CTLA-4 and anti-PD1 treatment leads to a higher incidence of toxic side effects compared to the use of either drug alone.

### 5.2. Characteristics of Intestine-Related irAES

The clinicopathological features of intestine-related irAES bear a striking resemblance to those observed in ulcerative colitis and Crohn’s disease. In clinical practice, cancer treatment typically involves a combination of tumor immunotherapy, radiotherapy, and chemotherapy. It is noteworthy that chemotherapy alone carries a certain risk of inducing colitis in patients, as agents like carboplatin and cisplatin exhibit inherent intestinal mucosal toxicity. A documented early clinical case [100] illustrated the development of severe and life-threatening IBD in a patient after undergoing four cycles of combined chemotherapy with paclitaxel and cisplatin. Consequently, the incidence of clinical irAEs involving the intestine appears to be substantially elevated.

During colonoscopy, ulcerative colitis is commonly characterized by the presence of blurred, disordered, or even absent vascular texture of the mucosa. Other noticeable features include mucosal congestion, edema, friability, bleeding, and purulent secretion. The mucosa typically appears rough and exhibits fine granulation. Diffuse, multiple erosions or ulcers may also be observed, along with shallow, blunt, or disappearing colonic pouches, pseudopolyps, and mucosal bridges [101].

In the case of ICB-associated IBD, endoscopic findings often include exudation of tissue fluid, loss of vascular morphology, mucosal granularity or edema, plaque or diffuse erythema, and diffuse ulcers. Previously, the treatment approach for ICB-related IBD closely resembled that of ulcerative colitis. Patients would discontinue tumor immune drugs and receive treatment with glucocorticoid prednisolone, which demonstrated certain efficacy [102]. However, it is important to note that glucocorticoids are immunosuppressive agents, which can significantly dampen the effectiveness of tumor immunotherapy.

## 6. Strategies to Mitigate Adverse Effects of Tumor Immunotherapy

Unlike the conventional pathogenesis of IBD, the primary etiology of ICB-related IBD is predominantly associated with the immune system. Various forms of ICB-related IBD can arise from different tumor immunotherapies. By effectively mitigating irAEs induced by ICB, including IBD, it becomes possible to sustain the administration of tumor immune drugs. This approach not only reduces the adverse effects of ICB but also significantly enhances the efficacy of tumor immunity.

### 6.1. Conventional Remission Therapy for irAEs

Currently, the clinical management of ICB-related IBD closely resembles that of conventional IBD. In addition to surgical interventions, pharmacological treatments for IBD can be categorized into non-biological agents and biological agents. Non-biological agents primarily encompass aminosalicylic acid agents, hormones, and thiopurine immunosuppressants. On the other hand, biologics approved for IBD treatment mainly consist of anti-TNF and anti-integrin agents. It is important to note that most of these medications possess immunosuppressive properties or promote the activity of immunosuppressive cells. While these immunosuppressive agents can alleviate ICB-associated IBD, they often facilitate the progression of cancer and are associated with various other side effects. Even the anti-TNF-α monoclonal antibody infliximab exhibits a substantial incidence of side effects.

### 6.2. Infliximab

In clinical practice and research, infliximab, an anti-TNF-α monoclonal antibody, is commonly employed to mitigate irAEs associated with IBD or liver disease. It is also widely utilized in the treatment of various autoimmune diseases and conventional IBD. In a study conducted by researchers [14], mouse models of colon cancer and melanoma were generated, and colitis was induced using low-dose DSS and ICIs with anti-CTLA4 and anti-PD1. The results of the study demonstrated that the administration of infliximab, as an anti-TNF-α agent, did not compromise the anti-tumor efficacy of ICIs. Instead, it led to increased infiltration of CTLs into tumors and alleviated colitis. These findings suggest that the resolution of irAEs can enhance the antitumor effects of ICIs. 

In another clinical treatment case [103], a patient with metastatic melanoma received ICI therapy but developed severe irAEs, experiencing over 18 episodes of diarrhea per day for two weeks, indicating severe colitis. However, discontinuation of tumor-immune drugs, along with the use of steroids and infliximab, did not significantly alleviate the irAEs. Subsequently, the medical team discovered that the patient had a severe intestinal viral infection, and the patient’s condition improved following the combined administration of antiviral drugs. These findings suggest that anti-TNF-α therapy may not be effective in alleviating irAEs associated with IBD.

Cancer is a chronic disease that often leads to immune cell exhaustion in patients [25]. Additionally, the complexity of the human intestinal flora far exceeds that of mice residing in specific pathogen-free (SPF) animal facilities. The use of DSS to induce irAEs in the study [14] represents a chemical corrosive injury model, which differs from the mechanisms underlying IBD induced by ICIs in humans. The treatment approach for human ICB-related IBD typically involves discontinuing the use of ICB drugs and initiating infliximab therapy, which inevitably impacts tumor growth. Infliximab, being an immunosuppressive agent, increases the risk of other infections in humans and downregulates the immune response of the body. Over time, the human body may activate alternative pathways to synthesize other types of tumor necrosis factor. While infliximab can currently be used clinically to alleviate irAEs, its long-term implications remain unclear. Moreover, the use of infliximab has been associated with side effects such as psoriasis [104] and even aseptic meningitis [105]. Therefore, the use of infliximab to manage irAEs does not resolve the inherent contradiction of suppressing the therapeutic effect of tumors.

### 6.3. Targets for Mitigating irAEs and Enhancing Tumor Immunotherapy

Currently, numerous treatment methods exist for IBD, but the majority of them rely on immunosuppression or the promotion of immunosuppressive cells. Consequently, while these treatment approaches may be suitable for patients with conventional IBD, they are not applicable to individuals with IBD associated with irAEs. Hence, there is a pressing need for strategies that can effectively manage irAEs without compromising tumor immunotherapy efficacy.

In a study [106], it was demonstrated that anti-CTLA-4 primarily activates CD4^+^ T cells, while anti-PD1/PDL1 primarily activates CTLs. Moreover, CD4^+^ T cells produce a greater quantity of inflammatory factors compared to CTLs. Based on these findings, alternative drugs can be considered in place of anti-CTLA-4 monoclonal antibodies, in combination with anti-PD1, to downregulate irAEs associated with IBD and potentially enhance the anti-tumor effects. For instance, a study [107] revealed that nifedipine (NIFE), a calcium channel blocker, possesses the ability to inhibit calcium influx and impede the function of the nuclear factor of activated T cells 2 (NFAT2). This inhibition prevents the transcriptional activation of downstream signaling molecules, leading to the inhibition of proliferation and metastasis of colorectal cancer. Furthermore, nifedipine has demonstrated the capacity to reduce the expression of PD-L1 in colorectal cancer cells and PD1 on CD8^+^ T cells. This activation of tumor immune surveillance enhances the efficacy of PD-1-based anti-tumor immunotherapy.

In another study [108], researchers explored the potential of combining alternative drugs with PD-1/PD-L1 treatment. By targeting the knockout of the c-Rel subunit of NF-κB specifically on Treg cells, in combination with anti-PD1 therapy, they achieved a reduction in the number of Treg cells within the TME. This approach resulted in a notable inhibition of melanoma growth, whereas the efficacy of PD1 treatment alone in wild-type mice was unsatisfactory. To target the ablation of the c-Rel protein, the researchers utilized pentoxifylline, an FDA-approved drug, and combined it with anti-PD1 treatment in melanoma-bearing wild-type mice. This combination approach demonstrated significant suppression of tumor growth. Therefore, utilizing pentoxifylline instead of anti-CTLA4 may hold the potential to downregulate irAEs and enhance the effectiveness of tumor immune responses.

Combining cell activation and metabolic checkpoint inhibition (CAMCI), as discussed in a study [109], involves the activation of targeted effector CD4^+^ T cells and memory CD4^+^ T cells, followed by their elimination using metabolic checkpoint inhibitors. This approach has shown promise in increasing the proportion of Treg cells and effectively treating immune-related IBD without downregulating the levels of inflammatory factors. While CAMCI may hold potential for application in IBD-associated irAEs, it is important to note that it also leads to an increased proportion of Treg cells, which can have detrimental effects on tumor immunity.

Instead of using anti-CTLA4, probiotics have emerged as a potential option to be combined with anti-PD-1 for tumor immunotherapy. A study [110] demonstrated that Bifidobacterium, a type of probiotic, can effectively reduce the severity of common IBD. Furthermore, another study [111] revealed that certain probiotics not only enhance tumor immunity but also alleviate the severity of ICB-related IBD. In this context, researchers discovered that a high-salt diet could suppress the expression of PD1 while promoting the enrichment of INFγ and hippurate, thereby elevating the levels of NK cells. Remarkably, the combination of a high-salt diet and anti-PD1 treatment resulted in significant tumor regression. However, high-salt diets are associated with numerous side effects. To address this, researchers found that the high-salt diet led to an increase in the abundance of Bifidobacterium, heightened hippurate enrichment, and enhanced anti-tumor effects when combined with anti-PD1 therapy. Subsequently, the researchers employed fecal microbiota transplantation (FMT) from the feces of healthy mice on a high-salt diet, combined with anti-PD1 treatment. This approach not only reduced side effects but also achieved tumor regression [111]. Another study [112] on FMT also suggests that fecal microbiota transplantation may be a promising avenue to combat tumors and mitigate irAEs in IBD.

In addition to T cells, neutrophils and tissue-resident Kupffer cells play a significant role in liver injury following ICI treatment. For instance, in a study conducted by researchers [13], a hepatic model of irAEs in the context of immune checkpoint blockade was developed. The study revealed that liver toxicity is mediated by tissue-resident Kupffer cells, which detect IFN-γ and subsequently produce IL-12/23. The combined action of IL-12 and IFN-γ induces neutrophil responses, and these neutrophils contribute to liver injury through the release of TNF. Therefore, targeting Kupffer cells is suggested as a potential approach to alleviating hepatic irAEs associated with immune checkpoint inhibitors.

## 7. Targeting Epithelial Cells May Represent a Crucial Approach for Modulating the Delicate Equilibrium between Tumor Immunotherapy and irAEs 

### 7.1. Immune Dysregulation Stands as a Significant Contributing Factor to ICI Colitis

A comprehensive investigation of immune cell populations in intestinal irAEs following ICI therapy was detailed in a study [41]. Mass spectrometry flow cytometry and single-cell sequencing were employed to assess changes in immune cell composition and cytokine profiles among melanoma patients receiving ICI treatment. Patients with ICI colitis who underwent combined anti-CTLA4 and anti-PD1 therapy exhibited an enrichment of bone marrow-derived cells, notably inflammatory DCs, alongside a reduction in CD4^+^ tissue-resident memory T cells. Moreover, there was an increase in Tregs, enrichment of CD8^+^ T cells within the lamina propria and mucosa, significant elevation of CTLs and circulating CD8^+^ T cells, and a notable decrease in intestinal protective IL-22-producing ILC2/3. Additionally, both ILC1 cells and all CD8^+^ T cells displayed an enrichment of GZMB. Remarkably, only circulating CD8^+^ T cells exhibited a strong enrichment of MKI67, a marker indicating cell proliferation. These findings suggest that the combined effect of IFN-γ and TNF-α production, along with the enrichment of GZMB^+^ CD8^+^ T cells, may contribute to intestinal tissue damage in irAEs associated with IBD. Furthermore, a significant decrease in the expression of IL-22-producing ILC2/3 cells within the intestinal tissue is identified as an additional factor underlying intestinal injury. IL-22, a cytokine that acts on intestinal stem cells (ISCs), plays a crucial role in their protection, nourishment, and preservation, especially for intestinal Lgr5^+^ stem cells [113] (Figure 3A).

### 7.2. Similarities between ICIs-Related IBD and GVHD-Related IBD

Typically, bone marrow transplantation represents the sole curative approach for leukemia. This procedure involves transplanting bone marrow hematopoietic stem cells, which closely match the patient’s HLA, after the eradication of bone marrow cells through intensive radiotherapy [116]. GVHD arises from a variety of lymphocyte-mediated inflammatory cytokine storms triggered by histocompatibility antigen disparities. The skin, mucosa, intestine, and liver are the organs most commonly affected, leading to symptoms such as bloody diarrhea, intestinal mucosal shedding, and intestinal obstruction. The overall incidence rate of GVHD can reach as high as 60% [117], with a mortality rate ranging from 5% to 20%. In a study [114], researchers observed significant increases in intestinal permeability and tissue TNF-α levels, as well as an enrichment of CD8^+^ T cells and the GZMB they secreted in GVHD (Figure 3B). Another investigation of intestinal GVHD [113] demonstrated that MHC-identical bone marrow transplantation (BMT) or transplantation with MHC-similar T cell-depleted bone marrow did not induce GVHD. In GVHD model mice, there was a significant decrease in the number of ILCs and the expression of IL-22 secreted by ILCs [113]. Additionally, a separate study [48] revealed that macrophage-released IL-23 inhibited the production of IL-22 by ILC3 cells, contributing to intestinal epithelial damage in colitis. By comparing ICIs-related IBD and GVHD-related IBD, it becomes apparent that both conditions share similarities in their molecular pathogenesis and clinical symptoms. Therefore, treatment strategies employed for GVHD-related IBD may hold potential for application in ICIs-related IBD as well.

### 7.3. MLCK Targeting for GVHD-Related IBD and ICIs-Related IBD: Promising but Imperfect

Long myosin light chain kinase (MLCK) is the predominant isoform expressed in the human intestinal epithelium, primarily responsible for regulating the permeability of tight junctions (TJs) in epithelial cells. In a previous in vivo study [118], researchers induced acute diarrhea in mice by injecting an antibody to activate CD3^+^ T cells with the aim of investigating the involvement of the intestinal epithelial barrier in the pathogenesis of diarrhea disease. Elevated MLCK expression resulted in the increased phosphorylation of MLC within the tight junctions and reorganization of F-actin around the junctions, enhanced the permeability of tight junctions, and facilitated the rearrangement of ZO-1 and occludin proteins. Additionally, MLCK upregulation promoted the endocytosis of occluding [119]. Conversely, inhibiting MLCK activity prevented these alterations in tight junction structure, thereby reversing the decreased permeability of the intestinal epithelium.

The MYLK1 gene encodes three types of MLCK proteins: long MLCK1/2 and short MLCK. These MLCK variants are expressed in various tissues and organs, including the myocardium, smooth muscle, bronchi, and vascular endothelium, among others. They share catalytic domains and calmodulin-binding regulatory domains. A complete knockout of MLCK in mice [120] leads to perinatal death. Similarly, the use of MLCK-targeted inhibitors such as ML-7, PIK, and others can result in weight loss and lethargy in mice. Furthermore, a specific knockout of the MYLK gene in smooth muscle tissue causes hypotension, bladder dysfunction, severe intestinal motility disorders, and even mortality [121]. Interestingly, in CA-MLCK mice with upregulated MLCK expression, there is a substantial increase in major histocompatibility MHC-I, CD4^+^ T cells, TNF-α, and IFN-γ in the intestinal mucosa, accompanied by increased intestinal permeability. In an experimental model of CD4^+^CD45^+^Rb^hi^-induced T cell transfer colitis, CA-MLCK mice exhibit more severe IBD symptoms [122].

Long MLCK1/2 is predominantly expressed in the intestinal epithelium, and among these, long MLCK1 is the isoform specifically regulated in the intestinal epithelium, distinct from long MLCK2 due to differences in their IgCAM3 domains. In models of chronic immune-mediated IBD, the activation of TNF leads to increased phosphorylation of MLC via MLCK1, while MLCK2 does not demonstrate such effects [123]. In active IBD cases, MLCK1 primarily localizes to the apical cells of intestinal epithelial cells and forms a discernible cell lineage on the perijunctional actomyosin ring (PAMR). Conversely, in healthy mice, MLCK1 exhibits a diffuse distribution in colon cells [123]. Subsequently, a library screening approach identified a small molecule called divertin, which selectively binds to the IgCAM3 domain of long MLCK1, leading to the inhibition of MLCK1 activity. Experimental findings demonstrated that divertin effectively mitigates symptoms of IBD in mice without inducing weight loss, systemic toxicity, or epithelial mucosal toxicity. However, the use of divertin significantly downregulates various inflammatory factors in the intestine. When considering the application of divertin for suppressing side effects in tumor immunotherapy, this may potentially have a detrimental impact on the efficacy of tumor immunotherapy.

Based on the aforementioned findings, researchers discovered [114] that upregulation of MLCK significantly amplified the severity of IBD associated with GVHD. However, when MLCK was specifically knocked out in the intestinal epithelium of villi, it led to a significant alleviation of GVHD-related IBD (Figure 3C). This intervention resulted in reduced intestinal permeability, restored body weight, increased survival rate, and notable downregulation of CD8 cells in the intestinal epithelium. Additionally, the expression of GZMB decreased (Figure 3B), and the tight junction protein ZO-1 exhibited recovery. ZO-1 is a crucial component for maintaining barrier function, facilitating effective mucosal repair, and promoting the proliferation of mucosal epithelial cells [124].

Indeed, in patients with GVHD, strategies that aim to reduce intestinal permeability and downregulate MLCK1 expression, as well as inflammatory factors, can greatly alleviate GVHD-related IBD and improve patient outcomes. However, it is important to note that this approach may not be ideal for patients with irAEs-related IBD (ICI colitis). This is because downregulating levels of TNF-α and IFN-γ, which are involved in the inflammatory response, can potentially compromise the efficacy of tumor immunotherapy. Balancing the management of IBD symptoms while preserving tumor immune responses is a complex challenge that requires careful consideration and personalized approaches for patients with ICI colitis.

### 7.4. Promising Regimen for Remission of ICIs Colitis

A recent study by [115] revealed that TNF-induced recruitment of MLCK1 to the PAMR is mediated by FKBP8. Peptidyl-prolyl cis-trans isomerase 8 (FKBP8) binds to MLCK1’s IgCAM3 domain through its PPI domain, facilitating its recruitment to PAMR. A targeted knockout of FKBP8 reversed MLCK1 enrichment at the cellular level, resulting in the downregulation of MLC phosphorylation and increased epithelial resistivity, leading to reduced cell permeability. In an immune-induced mouse model of IBD, specific inhibition of the FKBP family in intestinal tissue significantly reduced MLCK1 recruitment, phosphorylated MLC levels, intestinal permeability, and CD3^+^ T cell levels in the intestinal epithelium, without affecting TNF-α and IFN-γ levels [115]. It is noteworthy that elevated expression of MLCK1 has been observed in human Crohn’s disease tissues. This is accompanied by an increase in the interaction between MLCK1 and FKBP8, as well as a decrease in the levels of ZO1, which is one of the key proteins involved in maintaining the intestinal mucosal barrier function (Figure 3D,E). These findings suggest that targeting FKBP8 holds promise for alleviating intestinal IBD without downregulating TNF-α and IFN-γ levels. Furthermore, this approach, if applied to ICI colitis, may potentially reduce the impact on tumor immunotherapy and mitigate the side effects of immune checkpoint blockade.

## 8. Conclusions

This comprehensive review analyzes multiple therapeutic approaches for mitigating irAEs associated with tumor immunotherapy. However, nearly all methods examined in this study exhibit limitations, including compromised tumor immune response, increased toxicity, or implementation challenges. Nonetheless, our retrospective analysis reveals that targeting intestinal epithelial cells, a strategy that preserves tumor immunity while offering simplified administration and reduced toxicity, holds significant potential in the field of tumor immunotherapy.

## Figures and Tables

**Figure 1 ijms-25-10919-f001:**
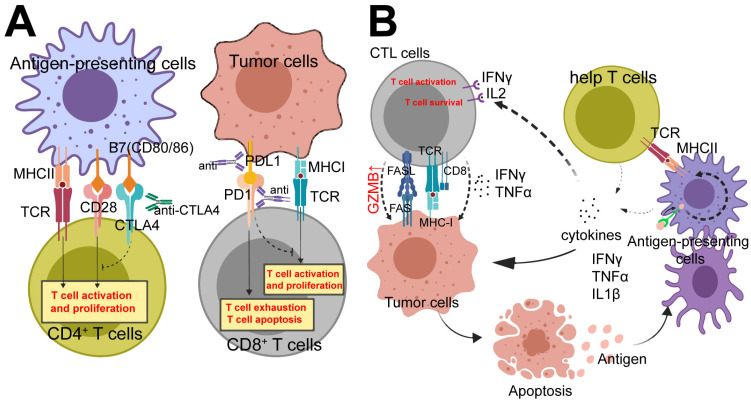
Immunocytes in Tumor Immunotherapy: (**A**) PD-1 and CTLA-4, functioning as immune checkpoints, exhibit potent inhibitory effects on anti-tumor immune cells. The blockade of immune checkpoint inhibitors (ICI) has shown the potential to enhance the capability of tumor immune cells. (**B**) Interactions among various effector immune cells during the process of tumor immunotherapy have been shown to enhance the efficacy of tumor immune responses.

**Figure 2 ijms-25-10919-f002:**
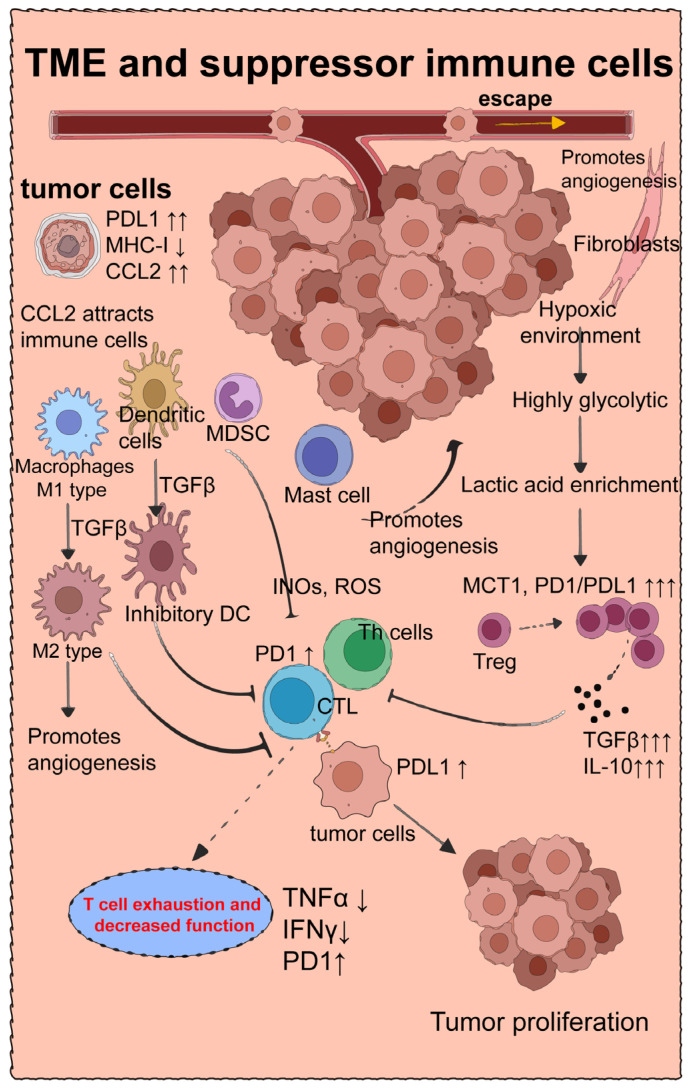
Tumor Microenvironment in Tumor Immunotherapy: The tumor microenvironment possesses distinctive physicochemical characteristics, capable of converting infiltrating immune cells into suppressive immune cells, thereby counteracting tumor immunotherapy and facilitating tumor growth and evasion.

**Figure 3 ijms-25-10919-f003:**
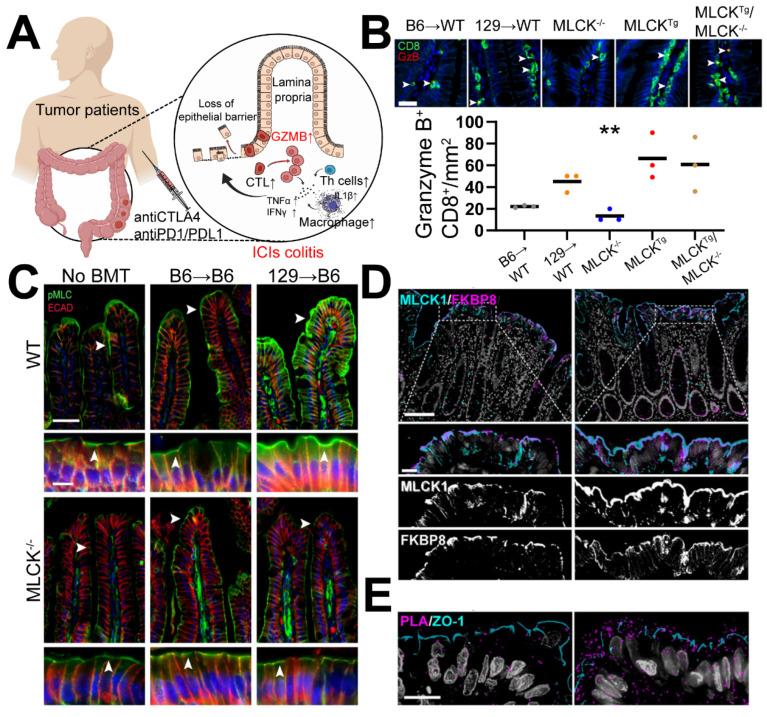
The Relationship between the Occurrence of ICI Colitis and Intestinal Epithelial Mucosal Barrier: (**A**) Simplified mechanisms underlying the occurrence of ICI colitis in tumor patients treated with combination anti-PD-1/PD-L1 and anti-CTLA-4 therapy. (**B**) Significant downregulation of GZMB levels in CD8+ cells upon MLCK knockout in a murine model of graft-versus-host disease [114]. (**C**) MLCK knockout significantly reduces phosphorylation levels of myosin light chain in intestinal epithelial mucosa of murine GVHD model [114]. (**D**) Remarkable upregulation of MLCK1 expression levels in patients with Crohn’s disease [115]. (**E**) Significant increase in the interaction level of MLCK1 with FKBP8, and downregulation of ZO1 expression in intestinal epithelial mucosa of Crohn’s disease patients: an immunofluorescence study [115].

## Data Availability

No datasets were generated or analyzed during the current study.

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
