# Peer review of "Balancing Tumor Immunotherapy and Immune-Related Adverse Events: Unveiling the Key Regulators"

_ijms, 2024, doi:10.3390/ijms252010919_

Round 1
Reviewer 1 Report
Comments and Suggestions for Authors
This is a well-written review article with a good organization profile/format to benefit readers. The Article is informative. Reference citation is in a good shape and cited many high impact and recent publications. Good job.
Specific comments are blow.
1. All abbreviations need to be defined from the first time of the full name coming out. For example, TME should be defined in parenthesis for the first time appearing as tumor microenvironment. Others like CICB, DSS, full name, etc. should be defined. Please check the entire review article.
2. Additionally, TME is a common abbreviation and should be used often after defined to save space. The author can define TME more than one time if needed (some examples authors did) but not TME and full name mixture. For example, under 4.3 still use full name of TME
3. The author just defined MDSC above in 4.3. So, no need to write “MDSC cells, short for myeloid-derived suppressor cells, can be classified into two …”. Please check the entire review manuscript to avoid unnecessary repeated definition or use of the full name of the defined abbreviations.
4. These authors may separate in the Figure 1C as a new figure. Thus, this will let authors conveniently put the new figure 1C (new figure 2) in the right position/place between Section 4.1 and 4.2 together with the complemental text description. In this case, please carefully to make sure the Figure 2 will be Figure 3, which will need to be changed all in the text.
5. “This work was supported by National Natural Science Foundation of China (82270589). This work was supported by National Natural Science Foundation of China (82070548). This work was supported by National Natural Science Foundation of China (81800464).” may be revised into This work was supported by three funds/grants from National Natural Science Foundation of China (82270589, 82070548 and 81800464) for succinct.
Author Response
Dear Esteemed Reviewer,
I have revised the manuscript as requested. The text in blue with strikethrough indicates deletions, while the text in red denotes additions or replacements. I appreciate your review of these changes.
Comments 1: All abbreviations need to be defined from the first time of the full name coming out. For example, TME should be defined in parenthesis for the first time appearing as tumor microenvironment. Others like CICB, DSS, full name, etc. should be defined. Please check the entire review article.
1.Dear Esteemed Reviewer,
Regarding your first comment, I have made the requested modifications. Please find the details attached. Comments 2: Additionally, TME is a common abbreviation and should be used often after defined to save space. The author can define TME more than one time if needed (some examples authors did) but not TME and full name mixture. For example, under 4.3 still use full name of TME 2.Dear Esteemed Reviewer,
In response to your second comment, I have made the necessary revisions throughout the manuscript as per your request. Please find the details in the attached document.
Comments 3. The author just defined MDSC above in 4.3. So, no need to write “MDSC cells, short for myeloid-derived suppressor cells, can be classified into two …”. Please check the entire review manuscript to avoid unnecessary repeated definition or use of the full name of the defined abbreviations.
3.Dear Esteemed Reviewer,
In response to your third comment, I have made the requested deletions and revisions in the original manuscript. I appreciate your attention to this matter and kindly ask you to review the changes.
Comment 4. These authors may separate in the Figure 1C as a new figure. Thus, this will let authors conveniently put the new figure 1C (new figure 2) in the right position/place between Section 4.1 and 4.2 together with the complemental text description. In this case, please carefully to make sure the Figure 2 will be Figure 3, which will need to be changed all in the text. 4.Dear Esteemed Reviewer,
In response to your fourth comment, I have revised the manuscript by changing Figure 1C to Figure 2, and renaming the original Figure 2 to Figure 3. I appreciate your review of these modifications.
Comments 5. “This work was supported by National Natural Science Foundation of China (82270589). This work was supported by National Natural Science Foundation of China (82070548). This work was supported by National Natural Science Foundation of China (81800464).” may be revised into This work was supported by three funds/grants from National Natural Science Foundation of China (82270589, 82070548 and 81800464) for succinct. 5.Dear Esteemed Reviewer,
In response to your fifth comment, I have made the requested deletions and modifications. Thank you for your valuable feedback.

Reviewer 2 Report
Comments and Suggestions for Authors
The authors of this manuscript, Jianshang Huang et al., have compiled valuable insights regarding immune-related adverse events (irAEs) in their review titled "Balancing Tumor Immunotherapy and Immune-Related Adverse Events: Unveiling the Key Regulators." While immunotherapy has significantly advanced cancer treatment, the potential benefits for patients are often diminished by the immunologic toxicities associated with these therapies. Therefore, understanding the key regulators of irAEs is crucial for optimizing immunotherapy outcomes. This review article effectively consolidates relevant information on this topic. However, the following comments need to be addressed regarding the accessibility of the information cited by the authors:
1. The introduction severely lacks references.
2. The sentence in lines 89 and 90 is not clear. Please rewrite for better clarity.
3. Paragraph form lines 113 to 118 is very repetitive. Please rewrite.
4. No reference cited for paragraph from lines 144 to 149.
5. References, 29 to 32 are are review works. Please cite original articles for that information in the text.
6. Sentence in the lines 425 to 427 does not have reference.
7. Reference 71 is a press release and does not have information cited in the text between lines, 439 and 440.
8. Reference 72 does not have the information cited in the text between lines 441 to 443. Please cite original work.
9. As mentioned in line 462, reference 75 is not a study but a review article. Please provide reference for the clinical epidemiological study.
Author Response
Dear Esteemed Reviewer,
First, I would like to sincerely apologize for not providing complete citations for the required references in my previous submission. I appreciate your comments and have made every effort to correct the erroneous citations and add any missing references as per your suggestions. I am truly grateful for your assistance in my academic journey; your generosity and commitment to thoroughness are greatly appreciated. Thank you for your valuable feedback.I have revised the manuscript as requested. The text in blue with strikethrough indicates deletions, while the text in red denotes additions or replacements. I appreciate your review of these changes.
Comments and reply:
1. The introduction severely lacks references.
Regarding your first comment, I have added the necessary references as you requested. I appreciate your review of these additions.
2. The sentence in lines 89 and 90 is not clear. Please rewrite for better clarity.
In response to your second comment, I have rewritten the original lines 89-90 as requested. Please refer to the revised document for details.
3. Paragraph form lines 113 to 118 is very repetitive. Please rewrite.
Regarding your third comment, your suggestion is very insightful. I have made the revisions as requested to enhance the clarity and precision of the sentences. Thank you for your guidance.
4. No reference cited for paragraph from lines 144 to 149.
Regarding your fourth comment, thank you for your reminder. I have added several references in sentences 152-158 of the revised manuscript. I greatly appreciate your suggestion.
5. References, 29 to 32 are are review works. Please cite original articles for that information in the text.
Regarding your fifth comment, your suggestion is very accurate. I have added and modified the referenced citations as requested. These changes have been made in sentences 161-190 of the revised document. Thank you for your insightful recommendations.
6. Sentence in the lines 425 to 427 does not have reference.
Regarding your sixth comment, the modifications and added references can be found in lines 439-441. Thanks
7. Reference 71 is a press release and does not have information cited in the text between lines, 439 and 440.
Regarding your seventh comment, thank you for your suggestion. I should not have directly cited the content from the review in the original reference 71. I have now found the original meta-analysis and properly cited it. You can find this citation in lines 452-454. Thank you for your guidance.
8. Reference 72 does not have the information cited in the text between lines 441 to 443. Please cite original work.
Regarding your eighth comment, thank you very much. I have located the original reference in lines 456-462 and made the necessary revisions. I appreciate your review of these changes.
9. As mentioned in line 462, reference 75 is not a study but a review article. Please provide reference for the clinical epidemiological study.
Regarding your ninth comment, your suggestion is indeed correct. The original reference 75 was only a review. I have revised it and located several original references, which I have included in lines 481-487. I appreciate your review of these changes, and I sincerely thank you for your valuable suggestions.
Finally, I would like to express my sincere gratitude for all your suggestions. Your input has significantly enhanced my review and helped reduce many errors. I truly appreciate your kindness, generosity, and meticulousness as a scientist. Thank you!
yours JianShang

Round 2
Reviewer 2 Report
Comments and Suggestions for Authors
The authors have addressed the reviewer's comments and no further edits are needed.